# Novel Non-Toxic Highly Antibacterial Chitosan/Fe(III)-Based Nanoparticles That Contain a Deferoxamine—Trojan Horse Ligands: Combined Synthetic and Biological Studies

Omar M. Khubiev [1], Victoria E. Esakova [1], Anton R. Egorov [1], Artsiom E. Bely [1], Roman A. Golubev [1,2], Maxim V. Tachaev [1], Anatoly A. Kirichuk [1], Nikolai N. Lobanov [1], Alexander G. Tskhovrebov [1] and Andreii S. Kritchenkov [1,2,*]

[1] Faculty of Science, Peoples' Friendship University of Russia (RUDN University), 117198 Moscow, Russia; ihubievomar1@gmail.com (O.M.K.); 1032192994@rudn.ru (V.E.E.); sab.icex@mail.ru (A.R.E.); borutobirama@gmail.com (A.E.B.); asdfdss.asdasf@yandex.ru (R.A.G.); tatchaev@mail.ru (M.V.T.); kirichuk-aa@rudn.ru (A.A.K.); nnlobanov@mail.ru (N.N.L.); alexander.tskhovrebov@gmail.com (A.G.T.)
[2] Metal Physics Laboratory, Institute of Technical Acoustics NAS of Belarus, 210009 Vitebsk, Belarus
* Correspondence: kritchenkov-as@rudn.ru

**Abstract:** In this study, we prepared chitosan/Fe(III)/deferoxamine nanoparticles with unimodal size distribution (hydrodynamic diameter ca. 250 nm, zeta potential ca. 32 mV). The elaborated nanoparticles are characterized by outstanding in vitro and in vivo antibacterial activity, which exceeds even that of commercial antibiotics ampicillin and gentamicin. Moreover, the nanoparticles are non-toxic. We found that the introduction of iron ions into the chitosan matrix increases the ability of the resulting nanoparticles to disrupt the integrity of the membranes of microorganisms in comparison with pure chitosan. The introduction of deferoxamine into the obtained nanoparticles sharply expands their effect of destruction the bacterial membrane. The obtained antibacterial nanoparticles are promising for further preclinical studies.

**Keywords:** chitosan; iron(III); deferoxamine; nanoparticles; antibacterial activity; toxicity





## 1. Introduction

Infectious diseases of bacterial etiology are the cause of a large number of deaths and disability among the population [1–3]. In addition, infectious diseases affect not only people, but plants and animals, thereby causing significant damage to agriculture [4–6]. Therefore, infectious diseases represent both a medical and economic problem. Antibiotics have been used to treat bacterial infections for decades [7]. Undoubtedly, the introduction of antibiotics into clinical practice was one of the most important milestones in the history of medicine [8,9].

However, the use of antibiotics is associated with three major problems. The first problem is associated with the general systemic toxicity of antibiotics, which causes side effects during treatment and requires significant restrictions, especially in the cases of elderly patients, pregnant women, and children [10–12]. The second problem is related environmental issues. Antibiotics are often detected in wastewater, food, and even food packaging. Therefore, they harm the environment [13–16]. The third problem is related to the emergence of antibiotic resistance in bacteria, and this requires the prescription of large doses of antibiotics and/or the simultaneous use of several antibiotics [17–19]. All these problems have stimulated an intensive search for alternatives to traditional antibiotics, and such studies are among the most important tasks of medicinal chemistry and pharmacology.

Chitosan is a non-toxic, biocompatible and biodegradable polymer [20], which belongs to the most important eco-friendly macromolecular compounds. In addition, chitosan itself exhibits moderate antibacterial activity. The antibacterial effect of some of chitosan

derivatives is comparable with that of common commercial antibiotics ampicillin and gentamicin, and its transfection activity is similar to that of commercially available vector lipofectamine. Meanwhile, the toxicity of these chitosan derivatives is much lower than that of the reference antibiotics or lipofectamine [21].

Iron(III) is one of the most non-toxic 3D transition metals in the periodic table [22,23]. Moreover, several non-toxic iron-based compounds with promising antibacterial activity (including in vivo) are described in the literature [24–27].

The non-toxic natural compound deferoxamine is part of a group of so-called siderophores [28,29]. These are small, high-affinity compounds that chelate iron. Siderophores are secreted by bacteria and help them to store iron. Antibacterial compounds conjugated to siderophores are actively taken up by bacteria and this is used in medicinal chemistry. Therefore, siderophores are often called the Trojan horse ligand [30,31].

In this study, we hypothesized that the non-covalent fusion of chitosan, iron(III) and deferoxamine (Trojan horse ligand) would lead to the formation of a novel antibacterial system. Such a system consists of natural non-toxic compounds and therefore should be a non-toxic and promising alternative to antibiotics. The results of the synthesis, characterization and investigation of the biological properties of this system are discussed in detail in the sections that follow below.

## 2. Materials and Methods

### 2.1. Materials

In this study, we used chitosan with a viscosity average molecular weight of 2.7 kDa and degree of acetylation of 10%, abbreviately named CH (Bioprogress, Losino-Petrovsky, Russia); iron(III) chloride hexahydrate and deferoxamine are abbreviately named DESF (Aldrich, St. Louis, MI, USA). All other chemicals and solvents were obtained from commercial sources and were used as received, without further purification.

### 2.2. Synthesis of Chitosan-Fe$^{3+}$ Complex (POX-1)

A chitosan sample (1.0 g) was dispersed in 50 mL of a 1% solution of acetic acid and was stirred (400 rpm) for 24 h at room temperature. Then, 10 mL of the resultant solution was diluted to 200 mL with distilled water (solution A). Next, 0.1 g of ferric chloride(III) hexahydrate was dissolved in 100 mL of water (solution B). Solutions A and B were mixed and stirred for 1 h. The resultant solution was freeze-dried to give POX-1 as a soft orange cotton-like material.

### 2.3. Synthesis of Chitosan-Fe$^{3+}$ Complex (POX-2 and POX-3)

First, 100 mg of POX-1 was dissolved in 100 mL of distilled water (solution C) and 100 mg of deferoxamine was dissolved in 100 mL of distilled water (solution D). Solutions C and D were mixed (solution E); one part of solution E was stirred for 2 h and freeze-dried to give POX-2, and another was stirred for 24 h and freeze-dried to give POX-3.

### 2.4. General Methods

The apparent hydrodynamic diameter and $\zeta$-potential of nanoparticles in water were estimated at room temperature (approximately 25 °C) using a Photocor Compact-Z instrument (Russia) at $\lambda = 659$ nm and $\theta = 90°$ (10 scans, each one for 15 s).

IR spectroscopy was recorded on a Shimadzu IRSpirit at 4700 to 350 cm$^{-1}$ (10 mg of sample without any specified sample preparation).

UV spectra were recorded using a Mettler UV5 spectrophotometer.

Differential thermal analysis (DTA) and thermogravimetric analysis (TGA) were performed on the SDT Q600 using a heating rate of 5 °C/min in the temperature range from 40 °C to 600 °C.

X-ray diffraction analysis was carried out on a Dron-7 X-ray diffractometer, using a $2\theta$ angle interval from 7° to 40° with scanning step $\Delta 2\theta = 0.02°$ and exposure of 7 s per point.

Cu $K_\alpha$ radiation (Ni filter) was used, which was subsequently decomposed into $K_{\alpha 1}$ and $K_{\alpha 2}$ components during the processing of the spectra.

Loading efficiency (LE) was calculated using the following equation:

$$LE = [(m(\text{deferoxamine total}) - m(c \text{ deferoxamine in supernatant}))/m(\text{deferoxamine total})] \times 100$$

The mass of deferoxamine in the supernatant was determined by UV spectroscopy at a wavelength of 252 nm (calibration curve method).

X-ray fluorescence analysis of the samples was performed on a Clever C-31 X-ray fluorescence spectrometer. The relative measurement error was ±7%. A rhodium tube with a voltage of 50 kV and a current of 100 μA acted as a generator of γ-rays. The samples were taken without filters for 2000 s.

High-resolution electrospray ionization mass spectrometry (positive ion mode) was carried out on a APEX-Qe ESI FT-ICR instrument (Bruker, Billerica, MA, USA) with $CH_3CN$ as a solvent.

Antibacterial activity (in vitro and in vivo) and toxicity were evaluated completely as previously described by some of our group [32–35].

## 3. Results and Discussion

### 3.1. Preparation of Nanoparticles POX-1, POX-2 and POX-3

Treatment of the chitosan solution with iron(III) chloride immediately results in the generation of yellow-colored nanoparticles POX-1 of unimodal size distribution with hydrodynamic diameter ca. 285 nm and a high positive zeta potential (ca. 32 mV, see Table 1). The formed nanoparticles do not change their characteristic size and zeta potential values in a water nanosuspension for at least 10 days. Remarkably, when immersed in water after lyophilization, POX-1 is almost instantly redispersed, with the complete restoration of the starting values of the hydrodynamic diameter and zeta potential.

**Table 1.** Characteristics of the obtained nanoparticles.

| Sample | D, nm | ζ, mV | Polydispersity Index |
|---|---|---|---|
| POX-1 | 285 ± 2 | +31.8 ± 0.1 | 0.11 ± 0.02 |
| POX-2 | 254 ± 1 | +32.0 ± 0.3 | 0.10 ± 0.02 |
| POX-3 | 260 ± 1 | +32.5 ± 0.2 | 0.11 ± 0.03 |

The addition of deferoxamine to the POX-1 nanosuspension leads to the rapid formation of novel POX-2 nanoparticles of smaller size (hydrodynamic diameter ca. 254 nm) and the same zeta potential value as for POX-1 (ca. +32 mV, see Table 1). In water, POX-2, after 24 h, converts into POX-3 with hydrodynamic diameter ca. 260 nm, while the zeta potential value remains unchanged. It is likely that POX-2 and POX-3 are the same system, but this will be discussed in the following sections. It should be noted that both POX-2 and POX-3 are completely redispersible after lyophilization.

To confirm that deferoxamine is part of POX-2 and POX-3 and is not present in the solution in free form, we separated the resulting POX-2 and POX-3 from the supernatant after mixing the POX-1 and deferoxamine solutions. High-resolution mass spectrometry with electrospray ionization of the supernatant did not reveal any deferoxamine signals. Therefore, deferoxamine is included in the POX-2 and POX-3 nanoparticles. The loading efficiency (LE) of deferoxamine was ca. 100%.

X-ray fluorescence analysis of POX-1, POX-2 and POX-3 confirmed the presence of iron in the samples.

POX-2 also was characterized by scanning electron microscopy. The SEM image of POX-2 is presented in Figure 1.

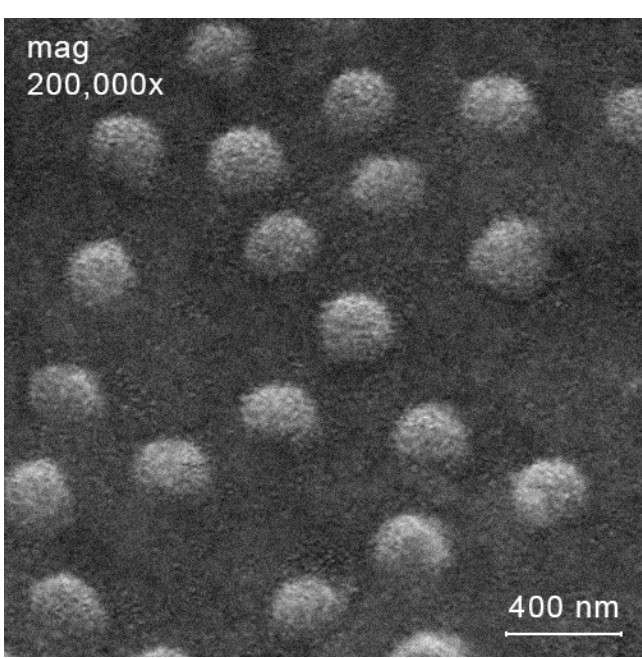

**Figure 1.** SEM image of POX-2.

### 3.2. Infrared Spectroscopy

The spectrum of $FeCl_3 \times 6H_2O$ (Figure 2A) displays small peaks of deformation vibrations of crystallized water at 1600 cm$^{-1}$ and stretching vibrations of free water at 3530 cm$^{-1}$. A wide double band of stretching vibrations of crystallized water exhibits two pronounced maxima at 3000 and 3220 cm$^{-1}$, while coordinated water deformation vibration bands arise at 840, 680, 600, 540 cm$^{-1}$. The spectrum of $FeCl_3 \times 6H_2O$ also displays stretching vibration bands attributed to Fe–Cl (360 cm$^{-1}$) and Fe–O (420 cm$^{-1}$) vibrations [36].

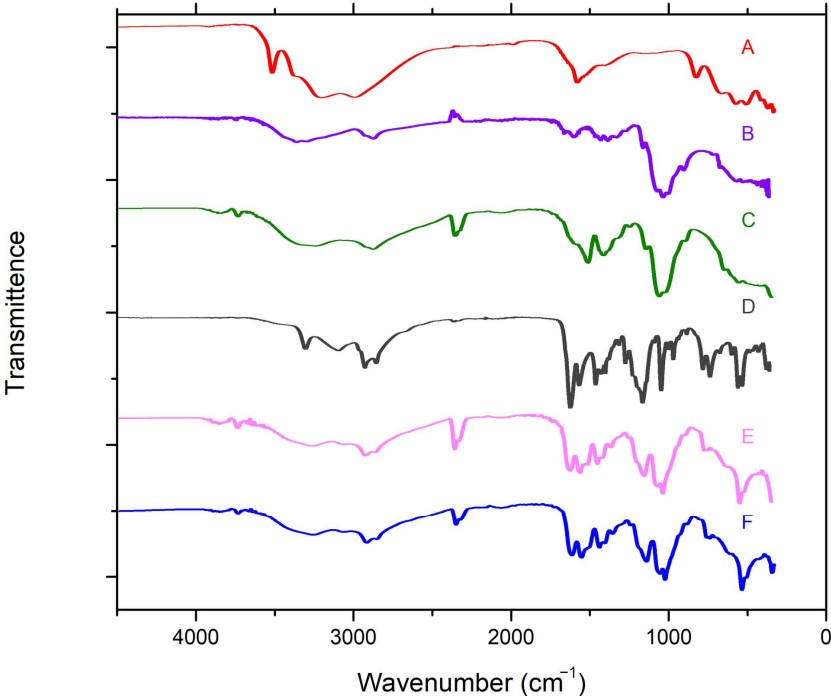

**Figure 2.** IR spectrum of $FeCl_3 \times 6H_2O$ (**A**); IR spectrum of chitosan (**B**); IR spectrum of POX-1 (**C**); IR spectrum of deferoxamine (**D**); IR spectrum of POX-3 (**E**); IR spectrum of POX-2 (**F**).

The spectrum of chitosan (Figure 2B) shows a wide band of O–H and N–H stretching ($3440–3100$ cm$^{-1}$), C–H stretching ($2870$ cm$^{-1}$) and bending ($1460, 1420, 1380$ cm$^{-1}$) vibrations, and N–H deformation vibrations ($1590$-$1650$ cm$^{-1}$). Absorption bands in the range of $900–1200$ cm$^{-1}$ are due to C–O–C, C–C and N–H deformation vibrations [37].

The spectrum of deferoxamine (Figure 2D) exhibits absorption bands of C=O, =O-H, -CONHR, -CH$_2$ stretching vibrations ($2840–3240$ cm$^{-1}$), peaks at $1620$ cm$^{-1}$ and $1560$ cm$^{-1}$ of stretching =N-CO and deformation NH vibration bands. Peaks of stretching vibrations -CH$_2$-CO are located in the interval $1400–1440$ cm$^{-1}$, while deformation aliphatic amine vibration peaks are found at $1030–1220$ cm$^{-1}$. In the range of $1040–500$ cm$^{-1}$, we observe deformation vibrations -(CH$_2$)$_x$-, -NH$_2$.

The spectrum of POX-1 (Figure 2C) displays the C–H stretching vibration band at cm$^{-1}$, which is shifted to $2890$ cm$^{-1}$ as compared to the starting chitosan. The spectrum shows a wide band of O–H and N–H stretching ($3440–3100$ cm$^{-1}$), C–H stretching ($2890–2870$ cm$^{-1}$) and bending ($1460, 1420, 1380$ cm$^{-1}$) vibrations and characteristic bands due to C–O–C and C–C deformation vibrations ($900–1200$ cm$^{-1}$). The N–H deformation vibration band is shifted to $1560–1520$ cm$^{-1}$ in comparison with the chitosan.

The spectrum of POX-2 (Figure 2E) exhibits stretching vibration bands characteristic of POX-1 (Figure 2C), i.e., O–H and N–H peaks ($3440–3100$ cm$^{-1}$) and a characteristic band at $2360$ cm$^{-1}$. C–H stretching vibration bands are located at $2920–2840$ cm$^{-1}$ and slightly shifted in comparison to POX-1. The spectrum of POX-2 exhibits stretching vibration bands characteristic of deferoxamine (Figure 2D), i.e., peaks of stretching =N–CO ($1620$ cm$^{-1}$) and deformation N–H ($1560$ cm$^{-1}$) vibration bands, stretching vibrations CH$_2$–CO ($1400–1440$ cm$^{-1}$) and a characteristic peak at $520–580$ cm$^{-1}$.

The spectrum of POX-2 (Figure 2E) is identical to the spectrum of POX-3 (Figure 2F). This confirms our assumptions about their identical structures.

### 3.3. X-Ray Diffraction

The spectra of chitosan and POX-1 are very similar. Both diffractograms show similar stretched peaks at $10–30°$ $2\theta$, which are attributed to chitosan (Figure 3A) [38].

The maxima of the spectra of both chitosan and deferoxamine are in the same region ($15–27°$ $2\theta$). The spectrum of chitosan displays a double wide peak with the left-shouldered maximum at $20°$ $2\theta$, while the diffractogram of deferoxamine at the same region exhibits four peaks with the main one at ca. $21.2°$ $2\theta$. Thus, the maximum and the shoulders of the deferoxamine-conditioned signals are right-shifted as compared to those of chitosan (Figure 3B).

The maxima of the spectra of both POX-2 and deferoxamine are located at ca. $21°$ $2\theta$. The spectrum of the POX-1 sample does not display any significant peak at this region. This fact indicates the presence of deferoxamine in POX-2 and its absence in POX-1 (Figure 3C).

The X-ray diffraction spectra of POX-2 and POX-3 samples are very similar. However, the diffractogram of POX-3 displays a broader maximum peak at $18–25°$. $2\theta$. This indicates the presence of smaller, crystallographically active structural units in the main particles of the sample (for example, in the POX-3 nanoparticles, probably, there are smaller nanolevel particles) (Figure 3D). The described differences do not refute our assumption about the identity of the chemical structure of POX-2 and POX-3.

Generally, the X-ray diffraction study confirms the identical chemical structures of POX-2 and POX-3, and allows us to conclude that both POX-2 and POX-3 contain chitosan and deferoxamine.

### 3.4. Differential Thermal Analysis (DTA) and Thermogravimetric Analysis (TGA)

The pure chitosan TGA curve has two stages of thermal degradation (Figures 4 and 5). The first stage is associated with a small weight loss due to the evaporation of the adsorbed and weakly bound water (mass loss 7%, $T_{max} = 121$ °C). This stage is accompanied by a weak endothermic effect. The second stage is associated with the degradation of the polymer structure (mass loss 89%, $T_{max} = 520$ °C).

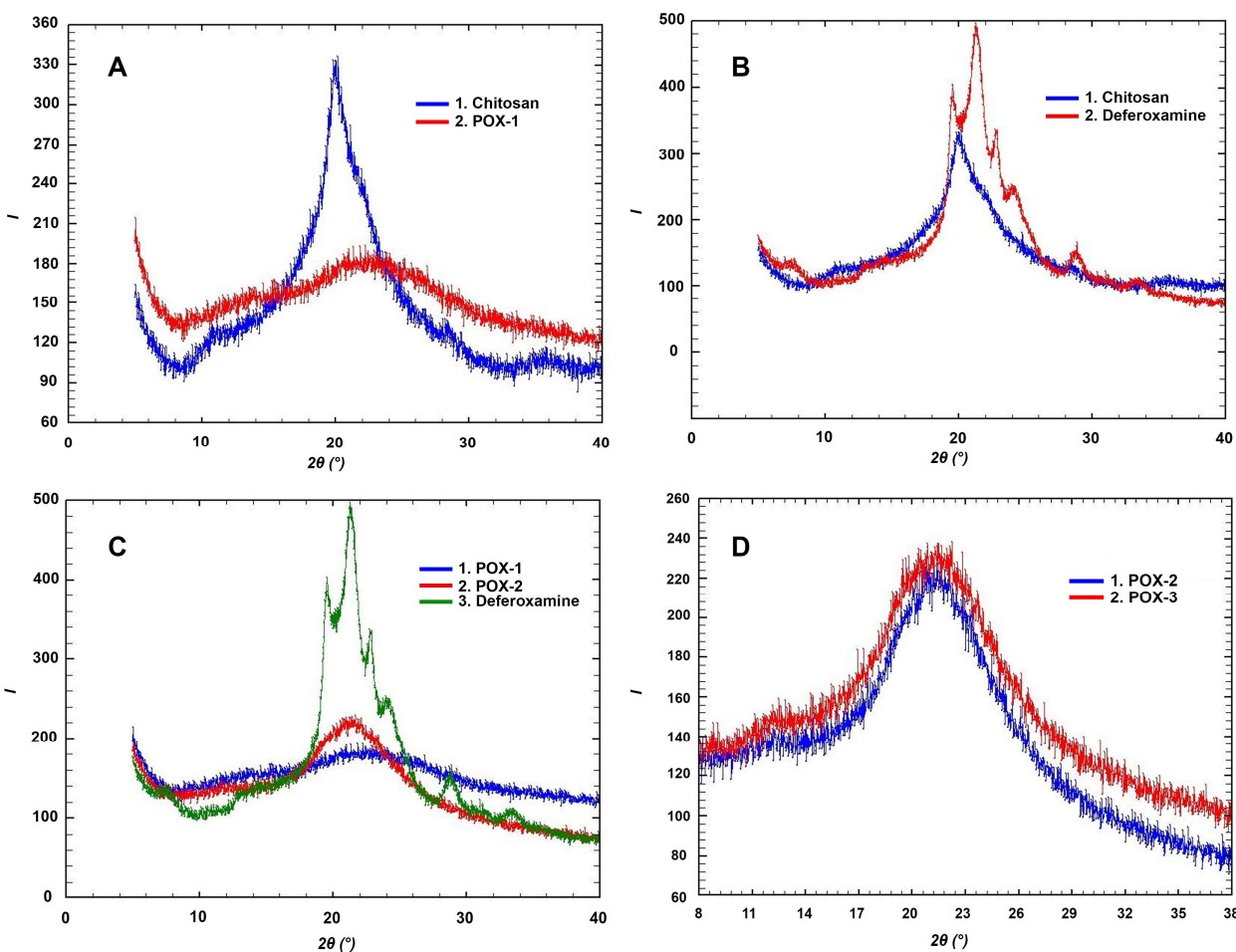

**Figure 3.** XRD patterns of chitosan and POX-1 (**A**); XRD patterns of chitosan and deferoxamine (**B**); XRD patterns of POX-1, POX-2 and deferoxamine (**C**); XRD patterns of POX-2 and POX-3 (**D**).

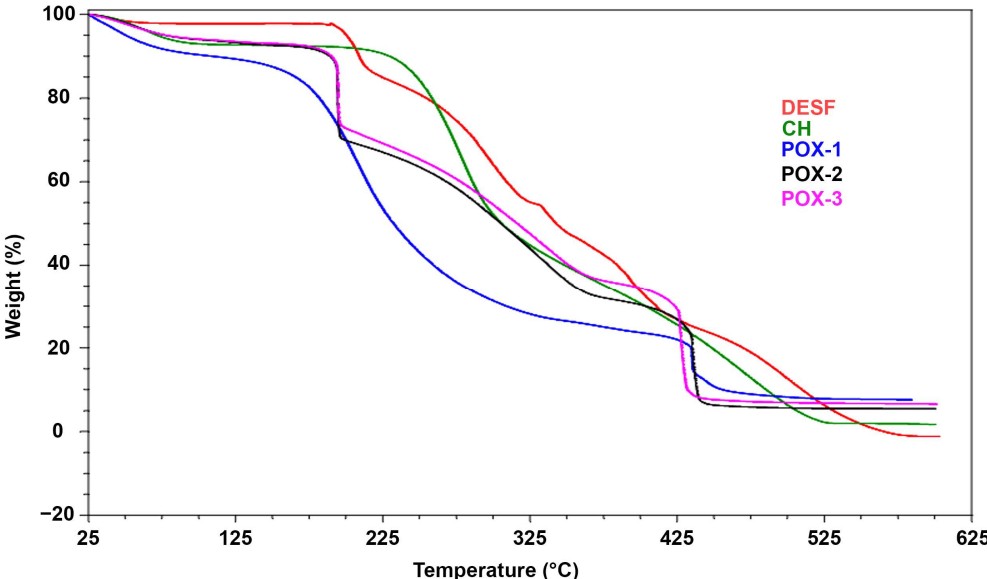

**Figure 4.** TGA curves of DESF, CH, POX-1, POX-2, POX-3.

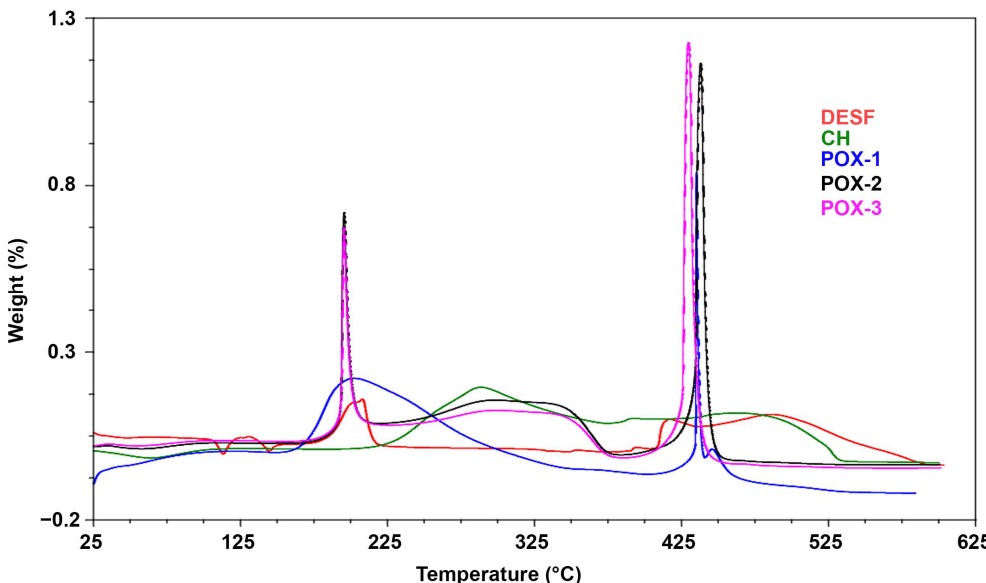

**Figure 5.** DTA curves of DESF, CH, POX-1, POX-2, POX-3.

POX-1 was obtained by the interaction of chitosan with $Fe^{3+}$ ions. The thermal degradation curve of POX-1 is characterized by three stages of degradation. The first stage is associated with water loss, and it is accompanied by a slight endothermic effect (mass loss 10%, $T_{max} = 122$ °C). The second stage is associated with the degradation of the polymer structure, and it is accompanied by a pronounced, unsharp exothermic effect (mass loss 73%, $T_{max} = 440$ °C). The third stage of decomposition has a spasmodic, pronounced exothermic effect (mass loss 10%, $T_{max} = 453$ °C) (Figures 4 and 5).

POX-2 and POX-3 were prepared via the interaction of POX-1 with deferoxamine. Their thermal decomposition curves are almost identical, and the values of weight loss and maximum temperature differ by no more than 10% (Figures 4 and 5). The first stage of thermal degradation is associated with water loss (mass loss 7%, $T_{max} = 146$ °C). The second stage is accompanied by an acute exothermic effect (mass loss 25%, $T_{max} = 200$ °C) followed by a gradual loss of mass. The third stage is accompanied by a sharp, pronounced exothermic effect ($T_{max} = 445$ °C, weight loss 20%). The POX-1 and POX-2 curves are significantly different from those of the starting deferoxamine and POX-1, which indicates the formation of new systems. The systems include the characteristic features of the thermal decomposition of both starting deferoxamine and POX-1: (i) deferoxamine-like weight loss at 200 °C (for deferoxamine, at 202 °C); (ii) POX-1-like sharp weight loss at 445 °C (for POX-1, at 453 °C). These observations are in agreement with the results of the X-ray diffraction study.

### 3.5. Biological Studies

3.5.1. In Vitro Antibacterial Activity

The in vitro antibacterial activity of the prepared nanoparticles was compared with that of the starting chitosan, iron(III) chloride hexahydrate and deferoxamine. For the in vitro evaluation of the antibacterial effect, we used the conventional agar well diffusion method. This method allows one to directly estimate the diameter of the microbial colonies' growth inhibition zone. The compound that provokes the largest zone of inhibition of bacterial growth is considered the most active antibacterial agent. The results of the experiments are presented in Table 2.

**Table 2.** Antibacterial effects of the elaborated nanoparticles.

| Sample | Inhibition Zone (mm) * | |
|---|---|---|
| | *S. aureus* | *E. coli* |
| Chitosan | 13.1 ± 0.1 | 9.7 ± 0.3 |
| Iron(III) chloride hexahydrate | 16.0 ± 0.3 | 12.2 ± 0.2 |
| Desferal | 2.6 ± 0.1 | 1.3 ± 0.1 |
| POX-1 | 22.4 ± 0.1 | 14.8 ± 0.2 |
| POX-2 | 29.7 ± 0.2 | 22.5 ± 0.1 |
| POX-3 | 29.9 ± 0.1 | 21.7 ± 0.3 |
| Ampicillin | 30.1 ± 0.3 | |
| Gentamicin | | 22.1 ± 0.1 |

* Mean value ± SD, n = 3.

Both the starting chitosan and iron(III) hexahydrate are characterized by moderate antibacterial activity, while their composite POX-1 is ca. 1.5 times more effective toward the tested bacteria. This can be explained by the formation of highly positively charged nanoparticles of POX-1. Our previous studies showed that, in many instances, chitosan-based nanoparticles exhibit much stronger antibacterial effects than the starting chitosan in its native form of a molecular coil [39].

Deferoxamine practically does not have an antibacterial effect. In contrast, the blending of deferoxamine with POX-1 results in POX-2 with outstanding antibacterial activity, which is comparable to that of the reference antibiotics ampicillin and gentamicin. We speculate that the high antibacterial activity of POX-2 is associated with the symbatic action of POX-1/deferoxamine in their complex with POX-2, but a fuller understanding of their mechanism requires additional biological studies.

The antibacterial effect of POX-3 essentially does not differ from that of POX-2.

The most effective antibacterial nanoparticles proved to be POX-2 and POX-3. The minimum inhibitory concentration (MIC) values were 0.14 µg/mL (*S. aureus*) and 0.19 µg/mL (*E. coli*) (compare with MIC values of ampicillin 0.18 µg/mL (*S. aureus*), gentamicin 0.23 µg/mL (*E. coli*)). Thus, the most active elaborated antibacterial nanoparticles, POX-2 and POX-3, are not inferior in their in vitro effect to the conventional antibiotics ampicillin and gentamicin.

3.5.2. Effect of the Integrity of the Bacterial Membrane

The main recognized model of the mechanism of the antibacterial effect of chitosan is associated with its polycationic nature [40]. Due to the protonation of primary amino functionalities, the neutral chitosan ($pK_a$ = 6.5) is converted into its polycation. The polycation interacts with the anionic moieties of the microbial cell, and this results in ionic pumps' dysfunction, osmotic imbalance, and overall membrane dysfunction, followed by cell membrane rupture and bacterial death [41].

To study the effects of the elaborated systems on the integrity of the microbial membrane, we used spectrophotometry of a suspension of bacterial cells in a 0.5% aqueous solution of sodium chloride in the UV region [42]. This approach is based on the fact that intracellular components are characterized by strong absorption at 260 nm [43]. In preliminary experiments, we found that both deferoxamine and iron(III) chloride did not damage the membranes of the tested bacteria. Thus, we compared the effect of starting chitosan and POX-1, POX-2 and POX-3 on the integrity of the cell membranes of *S. aureus* and *E. coli*. The results of the study are summarized in Figures 6 and 7.

In general, POX-1, POX-2 and POX-3 disrupt the integrity of the bacterial cell to a greater extent than the initial chitosan, and the effect of POX-2 and POX-3 is more pronounced than that of the starting chitosan. In addition, both in the case of *Staphylococcus aureus* and in the case of *E. coli*, POX-2 and POX-3 require less time to reach a plateau, i.e., their effect develops faster than that of chitosan of POX-1. Thus, the main mode of the

antibacterial effect of POX-2 and POX-3 is the destruction of the integrity of the bacterial cell membrane.

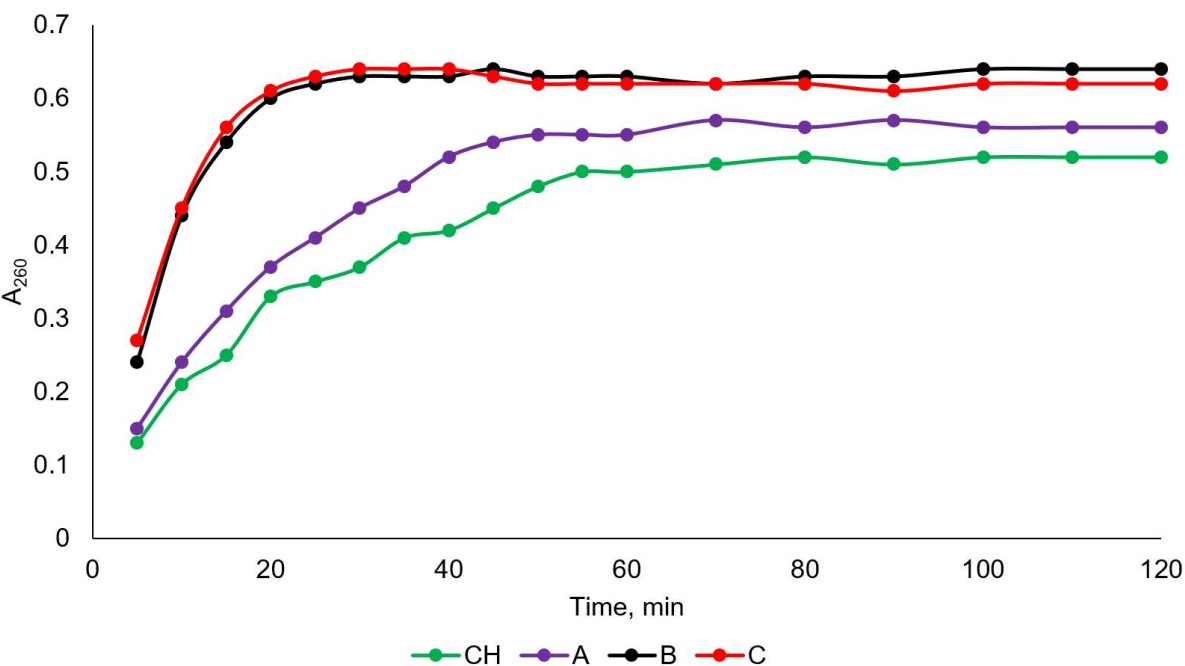

**Figure 6.** The effects of chitosan (CH), POX-1 (A), POX-2 (B), POX-3 (C) on the integrity of the cell membranes of *E. coli*.

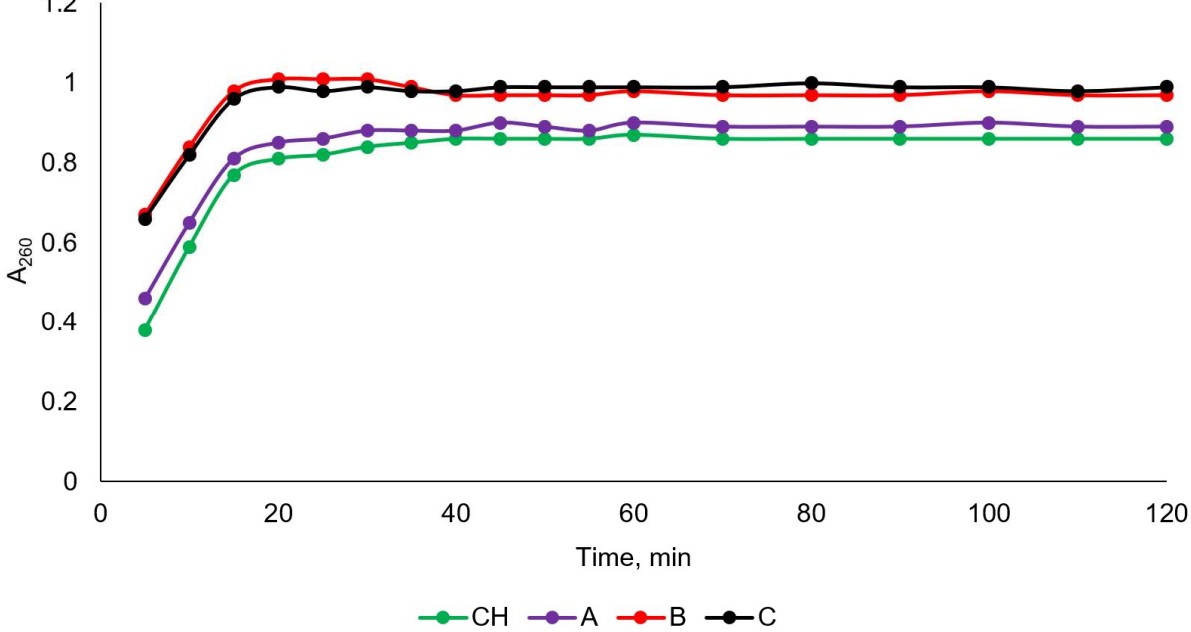

**Figure 7.** The effects of chitosan (CH), POX-1 (A), POX-2 (B), POX-3 (C) on the integrity of the cell membranes of *S. aureus*.

### 3.5.3. In Vitro Toxicity

The in vitro toxicity of the leading POX-2 and POX-3 was evaluated using the classic MTT test and compared with that of the starting chitosan, POX-1, deferoxamine and iron(III) chloride. As a quantitative measure of toxicity, we used the cell viability (CV, %) of the HEK-293 line after the cells were treated with a solution of the test substance (300 μg/mL).

The highest toxicity was found for iron(III) chloride (CV = 72%). The incorporation of iron(III) chloride into the chitosan polymeric matrix dramatically reduces its toxicity (CV = 94%). As a result, the formed nanoparticles POX-1 are characterized by the same toxicity as the starting chitosan (CV = 96%), which is considered a non-toxic polymer. The introduction of deferoxamine into POX-1 to give POX-2 or POX-3 did not lead to any noticeable changes in toxicity (for POX-1, CV = 93%; for POX-2, CV = 96%). However, it should be noted that the described nanoparticles are characterized by high positive values of the zeta potential; therefore, their intravenous administration into the general systemic circulation is undesirable, since it can cause the aggregation of negatively charged platelets. In this regard, in further experiments in vivo, we decided to use the intracavitary method of administration for these nanoparticles.

### 3.5.4. In Vivo Antibacterial Activity

At the next stage of the current work, we evaluated the in vivo antibacterial activity and toxicity of the leading systems, i.e., POX-2 and POX-3, in rats and compared their effects with those of antibiotics ampicillin and gentamicin.

The rats were subjected to the so-called model peritonitis. To imitate peritonitis, we infected the rats with a microbial mixture containing hospital strains of *S. aureus* and *E. coli*. Six hours after the introduction of microorganisms, all rats showed the conventional symptoms of peritonitis: lethargy, food refusal, rapid breathing and abdominal distention. In the control groups, exudate (200 µL) was collected with a sterile syringe 24 h after infection. A day later, all other rats were injected with a solution of the tested POX-2 or POX-3, or ampicillin or gentamicin. Then, 200 µL of exudate was taken after 7 h. To 200 µL of exudate, 1000 µL of 0.9% NaCl aqueous solution was added. Next, 100 µL of the resulting solution was evenly applied to a Petri dish with meat peptone agar. Colonies were counted 24 h after incubation at 37 °C. Subsequently, colony-forming units (CFU) were recalculated per 1 mL of exudate (Table 3).

**Table 3.** In vivo antibacterial effect.

| Tested Sample | CFU per 1 mL of Exudate (7 h after Treatment or 31 h after Infection) |
|---|---|
| Control without treatment (24 h after infection) | 2690 |
| POX-2 | 0 |
| POX-3 | 0 |
| Ampicillin | 540 |
| Gentamicin | 370 |

All tested samples demonstrated high in vivo antibacterial effects. However, POX-2 and POX-3 showed extremely high antimicrobial effects: no growth of colonies was found after collection of the exudate. The lower efficacy of the antibiotics can be explained by their rapid elimination after intracavitary administration. The elimination of polymer-based nanoparticles is much slower, and this leads to an increase in antibacterial activity compared low-molecular compounds, i.e., antibiotics, ampicillin and gentamicin. It should be noted that the results of this in vivo study in rats may not be directly applicable to human patients, and further studies, including clinical trials, are necessary to confirm the efficacy and safety of these nanoparticles in humans.

### 4. Conclusions

The results of this work can be considered from the following main perspectives.

Firstly, we elaborated the following types of chitosan-based nanoparticles: chitosan/Fe(III) (POX-1) and chitosan/Fe(III)/deferoxamine (POX-2 and POX-3, which differs from POX-2 in that it has been stirred in water for 24 h rather than 1 h). The resulting nanoparticles are characterized by close values of the hydrodynamic diameter and zeta potential. Characterization of the obtained POX-2 and POX-3 by physicochemical methods of analysis

clearly demonstrated that POX-2 and POX-3 are practically identical nanoparticles. The equal biological effects of POX-2 and POX-3 support this conclusion. Therefore, POX-2 is stable in water for at least 24 h, retaining all the characteristic features of its structure and biological effects.

Secondly, we evaluated the in vitro and in vivo antibacterial activity of the prepared POX-1, POX-2 and POX-3. The most effective antibacterial species are POX-2 and POX-3 and their antimicrobial efficiency is equal, which is not surprising since POX-2 and POX-3 are the same system. POX-2 and POX-3 are characterized by outstanding in vivo antibacterial effects, and their activity exceeds even that of commercial antibiotics ampicillin and gentamicin. Moreover, POX-2 and POX-3 are non-toxic.

Thirdly, we concluded that the introduction of iron ions into the chitosan matrix increases the ability of the resulting nanoparticles to disrupt the integrity of the membranes of microorganisms in comparison with pure chitosan. The introduction of deferoxamine into the obtained nanoparticles sharply expands their effect of destruction of the bacterial membrane.

Finally, we provide a fast, simple and convenient route to obtain highly effective, non-toxic antibacterial systems. This route is based on non-covalent chemistry and does not require laborious and sophisticated organic synthesis methods. The obtained antibacterial nanoparticles are promising for further preclinical studies, and this project is underway in our group.

**Author Contributions:** Conceptualization, O.M.K. and A.S.K.; methodology, O.M.K. and A.R.E.; software, A.E.B.; validation, A.A.K., A.G.T. and A.S.K.; formal analysis, N.N.L.; investigation, O.M.K. and V.E.E.; resources, R.A.G.; data curation, V.E.E.; writing—original draft preparation, A.R.E. and O.M.K.; writing—review and editing, A.G.T.; visualization, M.V.T.; supervision, A.S.K.; project administration, A.S.K.; funding acquisition, A.R.E. All authors have read and agreed to the published version of the manuscript.

**Funding:** This study was supported by the Russian Science Foundation, grant № 23-23-00021.

**Institutional Review Board Statement:** The animal study protocol was approved by the Ethics Committee and followed the recommendations of European Directive 2010/63/EU of 22 September 2010.

**Data Availability Statement:** Not applicable.

**Conflicts of Interest:** The authors declare no conflict of interest.

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
