# Peer review of "Novel Non-Toxic Highly Antibacterial Chitosan/Fe(III)-Based Nanoparticles That Contain a Deferoxamine—Trojan Horse Ligands: Combined Synthetic and Biological Studies"

_processes, doi:10.3390/pr11030870_

Round 1

Reviewer 1 Report

This study presents promising results regarding the development of chitosan/Fe(III)/deferoxamine nanoparticles with strong antibacterial activity, surpassing even that of commercial antibiotics like ampicillin and gentamicin. The use of iron ions and deferoxamine in the nanoparticles' composition increased their efficacy in disrupting the microorganisms' membrane and, consequently, their antibacterial activity. But some major modifications need before accepted for publication.

1-    I would recommend that the authors include more detailed information on the characterization of the prepared nanoparticles. This would help readers to better understand the reproducibility and validity of the study.

2-    It is important to note that the results of this study in rats may not be directly applicable to human patients, and further studies, including clinical trials, will be necessary to confirm the efficacy and safety of these nanoparticles in humans.

3-    It would be helpful to include more information on the potential mechanisms of toxicity of the tested systems, as this could provide insights into their potential use in clinical settings.

Author Response

1-    I would recommend that the authors include more detailed information on the characterization of the prepared nanoparticles. This would help readers to better understand the reproducibility and validity of the study.

  • In our opinion, nanoparticles are characterized by a very wide range of physical and chemical methods of analysis (IR spectroscopy, thermogravimetric analysis, X-ray analysis, X-ray fluorescence analysis, SEM, and, most importantly, hydrodynamic diameter and zeta potential. The synthesis of nanoparticles is described in detail. In addition, the synthesis is very simple and can be reproduced by first-year students (we used this technique in pedagogical work with first year students. Most importantly, if a researcher uses chitosan with the specified degree of deacetylation and molecular weight, he will get the same results. Thus, additional characterization of nanoparticles is unnecessary and will not solve the issue of reproducibility of the synthesis. Reproducibility depends only on the criteria of starting chitosan (molecular weight and degree of acetylation). We will be very grateful to Reviewer if he agrees with our arguments and facts.

2-    It is important to note that the results of this study in rats may not be directly applicable to human patients, and further studies, including clinical trials, will be necessary to confirm the efficacy and safety of these nanoparticles in humans.

  • Corrected.

3-    It would be helpful to include more information on the potential mechanisms of toxicity of the tested systems, as this could provide insights into their potential use in clinical settings.

  • Corrected.

Author Response

1. In page 2 of 15: within section 2.4. General methods: "The apparent hydrodynamic diameter and ζ-potential of nanoparticles in water was estimated at room temperature (about 25 °C) using a Photocor Compact-Z instrument (Russia) at λ=659 nm and θ=90°.″ 

Authors should notice the number of measurements performed in dynamic light scattering as well as the electrophoretic light scattering experiments.

  • Corrected.

2. In IR spectroscopy authors should describe in short the preparation of sample in IR spectroscopy measurements

  • Corrected.

3. On page 3 of 15: within the section 3.1.Preparation of nanoparticles POX-1, POX-2, and POX-3: ″ High-resolution mass spectrometry with electrospray ionization of the supernatant did not reveal any deferoxamine signals.″

Authors should calculated and quantitatively describe the loading efficacy of deferoxamine in nanoparticles.

  • Corrected.

4. On page 3 of 15: ″POX-2 also wascharacterized by scanning electron microscopy. The SEM image of POX-2 is presented in Figure 1. Authors should show all synthesized nanoparticles, POX-1, POX-2 and POX-3 by SEM. Additionally, they should show the size distribution of nanoparticles imaged by SEM in order to compare obtained size distribution with hydrodynamic diameter in suspension.

  • Unfortunately, we have only one SEM apparatus at our disposal. Even to get the SEM image shown in the manuscript, we had to wait more than a month in line. We will be very grateful to the Reviewer if he understands our difficult situation.

5. On page 4 of 15: table 1. The sign of zeta potential values of investigated nanoparticles should be included in Table 1.

  • Corrected.

6. Figures 3., 4., 5. and 6. could be summarized in one with A, B, C and D part. 

  • Corrected.

7. On page 9 of 15: "In Figures 7. and 8. abscises are  presented on time scale. Instead time-scale, temperature- scale should be state. 

  • Corrected.

8. After Abstract, the Abbreviation list sholud be included in the paper.

  • We revised the text of the manuscript and deciphered the abbreviations in the text of the manuscript.

Round 2

Reviewer 1 Report

The authors addressed all corrections commented on by the reviewer. Therefore, I highly recommend accepting and publishing this article in the current version.